# Antimicrobial and Antibiofilm Effect of Commonly Used Disinfectants on *Salmonella* Infantis Isolates

**DOI:** 10.3390/microorganisms11020301

**Published:** 2023-01-23

**Authors:** Katja Bezek, Jana Avberšek, Olga Zorman Rojs, Darja Barlič-Maganja

**Affiliations:** 1Faculty of Health Sciences, University of Primorska, 6310 Izola, Slovenia; 2Veterinary Faculty, Institute of Microbiology and Parasitology, University of Ljubljana, 1000 Ljubljana, Slovenia; 3Veterinary Faculty, Institute of Poultry, Birds, Small Mammals, and Reptiles, University of Ljubljana, 1000 Ljubljana, Slovenia

**Keywords:** *Salmonella* Infantis, biofilm, antibiofilm, disinfectants

## Abstract

*Salmonella enterica* subsp. *enterica* serovar Infantis is the most prevalent serovar in broilers and broiler meat in the European Union. The aim of our study was to test the biofilm formation and antimicrobial effect of disinfectants on genetically characterized *S.* Infantis isolates from poultry, food, and humans. For the biofilm formation under various temperature conditions (8 °C, 20 °C, and 28 °C) and incubation times (72 h and 168 h), the crystal violet staining method was used. The evaluation of the in vitro antimicrobial effect of Ecocid^®^ S, ethanol, and hydrogen peroxide was determined using the broth microdilution method. The antibiofilm effect of subinhibitory concentration (1/8 MIC) of disinfectants was then tested on *S.* Infantis 323/19 strain that had the highest biofilm formation potential. Our results showed that the biofilm formation was strain-specific; however, it was higher at 20 °C and prolonged incubation time. Moreover, strains carrying a pESI plasmid showed higher biofilm formation potential. The antibiofilm potential of disinfectants on *S.* Infantis 323/19 strain at 20 °C was effective after a shorter incubation time. As shown in our study, more effective precautionary measures should be implemented to ensure biofilm prevention and removal in order to control the *S.* Infantis occurrence.

## 1. Introduction

*Salmonella enterica* subsp. *enterica* serovar Infantis (*S.* Infantis) is the most prevalent serovar in broilers and broiler meat in the European Union (EU) and is among the top five serovars involved in human infections [1]. There is also an emerging problem of multidrug-resistant (MDR) *Salmonella* spp. isolates from broiler carcasses, which were reported with the highest proportion in Slovenia (90.9%) and Austria (87.3%), due mainly to serovar Infantis [2]. In addition, *S.* Infantis clones with a megaplasmid, known as plasmid of emerging *S.* Infantis (pESI), became predominant in Slovenian broilers and humans in the last decade [3]. pESI-like plasmids are highly mosaic and associated with MDR and increased virulence. In addition to several antimicrobial resistance genes encoded in pESI insertion regions, genes involved in resistance to quaternary ammonium compounds and heavy metals could also be present [4,5]. The presence of pESI-like plasmid was frequently found to be associated with an increased *S.* Infantis fitness under various environmental conditions [6,7]. As shown before, the strains harbouring the plasmid exhibit superior biofilm formation, adhesion, and invasion into avian and mammalian host cells [4]. Biofilm formation is often reported as one of the most important features of *Salmonella* persistence in the environment [8,9,10]. As pointed out by Merino et al. [11], *Salmonella* biofilms contribute to cross-contamination and foodborne infection thusly. It is, therefore, important to identify their presence, whereas different approaches are necessary for biofilm control [11]. Moreover, it has been shown that biofilms of several *Salmonella* isolates were more resistant to disinfectants [12] and antibiotics [13] than their planktonic counterparts. The resistance to disinfectants was shown to be higher also for *S*. Infantis strains which persist on farms [14]. Accordingly, there are difficulties of *S.* Infantis elimination from farms or slaughterhouses despite extensive cleaning and disinfection [15]. In addition, *S*. Infantis was repeatedly detected in certain holdings during the production break, although sanitation measures were implemented [16]. Whole-genome sequencing (WGS) of multiple isolates originating from the same farms also revealed the persistence of genetically closely related isolates; however, re-introduction of additional clones was also noted over time [3].

Disinfectants play an essential role in limiting the spread of infection and disease. However, the effectiveness of disinfection can be achieved only if an appropriate disinfectant is used at the right concentration [17]. In addition, all other biosecurity measures, such as cleaning, insect and pest control, and all in/all out systems, have to be implemented [18]. The cleaning and disinfection protocol commonly used in Slovenian broiler farms after transporting the birds to the slaughterhouse includes dry cleaning, i.e., removal of manure and feed, followed by soaking and washing of equipment and facilities, and disinfection. The most used disinfectants in the Slovenian poultry industry are peroxides combined with organic acids, a combination of glutaraldehyde and quaternary ammonium compounds, or a mixture of glutaraldehyde and formaldehyde. Recently, mixtures of hydrogen peroxide, acetic acid and peracetic acid, and chlorocresol products have also been introduced in some premises (personal communication). As reported previously, the slaughterhouse has also been identified as a potential source for *Salmonella* contamination of poultry meat, thus representing a critical stage for control of its dissemination in the food chain [19]. However, even when cleaning procedures were assessed as satisfactory and strong disinfectants were used, the survival and detection of *Salmonella* in carcass samples was reported [20]. Furthermore, contaminated surfaces in food processing environment may result in biofilm formation that can hardly be eliminated and thereby poses a risk of food contamination [21].

As is well known, the eradication of *S.* Infantis in poultry flocks and food processing chain is considered extremely difficult. In addition, due to its frequency of isolation from humans, *S.* Infantis is also an important public health concern. The aim of our study was, therefore, to test the biofilm formation, antimicrobial, antiadhesion, and antibiofilm effect of commercially used disinfectants on genetically characterized *S.* Infantis isolates from poultry, food, and humans.

## 2. Materials and Methods

### 2.1. Selection and Characteristics of S. Infantis Isolates

*S.* Infantis isolates originating from poultry (*n* = 7) and food (*n* = 3) were selected from the *Salmonella* strain collection of Veterinary Faculty, Institute of Microbiology and Parasitology, whereas clinical human isolates (*n* = 5) were provided by Mateja Pirš (Institute of Microbiology and Immunology, Faculty of Medicine, Slovenia). Isolates were previously characterised by whole-genome sequencing [3], and main characteristics are summarized in Table 1. pESI-positive strains with the Slovenian prototype plasmid (*n* = 8) harboured the typical pESI-associated resistance genes *aadA1*, *sul1*, and *tet*(A), which were absent in the strain with the truncated version of the plasmid. Additionally, all isolates belonged to sequence type 32 (ST32), and pESI-positive isolates (except 251/12 strain with the truncated version of the pESI plasmid) carried genes involved in resistance to quaternary ammonium compounds (*qacE*Δ*1*) and mercury (*mer*), whereas none had *ars* genes, involved in arsenic tolerance. 

### 2.2. Determination of Microbial Resistance to Disinfectants

Prior to experiments, the strains were subcultured on Trypton Soya Agar (TSA, Sigma-Aldrich, Darmstadt, Germany) aerobically at 37 °C. For inoculum preparation, 5 mL of TSB was inoculated with one bacteria colony and incubated overnight at 37 °C and shaken (60 rpm). The overnight cultures were then diluted in the ratio 1:100 to the final bacterial concentration in the experiment, which was approximately 1 × 10^7^ CFU/mL.

To determine the in vitro antimicrobial effect of Ecocid^®^ S (a mixture of potassium peroxysulphate, surfactant, organic acids, and inorganic buffer; Krka, d.d., Novo Mesto, Slovenia), ethanol (Sigma-Aldrich, Darmstadt, Germany), and hydrogen peroxide (Sigma-Aldrich, Darmstadt, Germany), a modified broth microdilution method was used [22,23]. Briefly, microdilution susceptibility testing was performed in flat-bottom 96-well clear plates (TPP Techno Plastic Products AG, Trasadingen, Switzerland) containing 100 μL of Trypton Soya Broth (TSB, Sigma-Aldrich, Darmstadt, Germany) in each column except the first one. The addition of disinfectant (100 μL) to the first two columns was followed by the twofold serial dilution across the plate. Each well was then inoculated with 20 μL of prepared bacterial culture. Plates were incubated aerobically at 37 °C for 24 h. After the incubation, bacterial viability was determined using PrestoBlue™ Cell Viability Reagent (Life Technologies, Darmstadt, Germany) according to the manufacturer’s protocol. After 90 min, the fluorescence signal was read using microplate reader (Infinite f200, Tecan Trading AG, Männedorf, Switzerland). The MICs were defined as the lowest concentration of the disinfectant at which no metabolic activity was detected. All the MIC measurements were carried out in three technical and two biological replicates. The control wells were prepared with a culture medium, with the bacterial suspension only, or alternatively, with the disinfectant dilutions only.

### 2.3. Biofilm-Forming Ability

For the biofilm formation under various growth conditions, the crystal violet (CV) staining method was used. Briefly, eight wells of a 96-well polystyrene flat bottom microtiter plate (TPP Techno Plastic Products AG, Trasadingen, Switzerland) were inoculated with 200 µL of prepared bacterial suspension and incubated under various temperatures (8 °C, 20 °C, and 28 °C) and incubation times (72 h and 168 h). After incubation, the suspensions were aspirated and the wells were washed twice with sterile phosphate buffered saline (PBS; Oxoid, Hampshire, UK). Plates were then dried at 60 °C for 10 min and stained adding 200 μL of 1% CV solution (Merck KGaA, Darmstadt, Germany) for 15 min. After that, the stain was aspirated, and the wells were washed five times under tap water and finally dried at 60 °C for 10 min. The bounded CV was then released by adding 200 μL of 96% ethanol (Sigma-Aldrich Co., St. Louis, MO., USA), and the optical density of distaining dye solution was measured at 595 nm using a microtiter plate reader (Infinite f200, Tecan Trading AG, Männedorf, Switzerland). The measurements were corrected by subtracting the mean values of the negative controls for each well (OD_595_) and discussed in terms of biofilm formation.

For the comparison of biofilm-forming ability at 20 °C and 28 °C after 72 h and 168 h, the isolates were classified as no biofilm, weak, moderate, or strong biofilm producers, as previously described by Stepanovic et al. (2000) [24]. Briefly, the cut-off optical density value (ODc) was three standard deviations above the mean OD of negative controls. Isolates were classified as OD ≤ ODc = no biofilm, ODc < OD ≤ 2× ODc = weak biofilm, 2× ODc < OD ≤ 4× ODc = moderate biofilm, and 4× ODc < OD = strong biofilm. 

#### 2.3.1. The Antiadhesion Assay

To quantify the antiadhesion properties of disinfectants, subinhibitory concentrations (1/8 MIC) were tested on *S.* Infantis 323/19 strain that had the highest biofilm formation potential. Briefly, the suspension of disinfectant dilutions and overnight bacterial culture were prepared. Eight wells of a 96-well polystyrene flat bottom microtiter plate (TPP Techno Plastic Products AG, Trasadingen, Switzerland) were then inoculated with 200 μL of prepared suspension and incubated for 24 h, 72 h, and 168 h at 20 °C in aerobic conditions. The positive control contained overnight bacterial culture diluted in sterile TSB, and the negative controls contained disinfectant dilutions without bacterial culture. After incubation, the assay followed the steps described for biofilm formation in Section 2.3. The measurements were corrected by subtracting the mean values of the negative controls for each well (OD_595_) and discussed in terms of antiadhesion effect.

#### 2.3.2. S. Infantis Biofilm Treatment with Disinfectants

The antibiofilm effect of disinfectants was tested on 72-h- and 168-h-old *S.* Infantis 323/19 biofilm formed on polystyrene at 20 °C (Section 2.3). For the biofilm treatment, disinfectants were prepared in PBS to final concentrations as used for surface disinfection as follows: 1% Ecocid^®^ S, 70% ethanol, and 1% hydrogen peroxide. All solutions were prepared at the time of testing. After biofilm formation, the bacterial suspensions were aspirated, and polystyrene plates were washed three times with sterile PBS and treated with disinfectants in prepared concentrations for 15 min and 30 min contact time. After the treatment, each well was washed again three times, and the numbers of biofilm cells were determined as CFU/mL after 10 min treatment in an ultrasonic bath (room temperature, 10 min; frequency, 37 kHz; power, 80 W) (Elmasonic P 60 H, Elma Schmidbauer GmbH, Singen, Germany). Then, the number of viable bacterial cells was determined by drop plate method on TSA. The experiment was carried out in three technical and two biological replicates. The difference in the reduction of viable bacterial cells after the treatments, compared with the untreated control, is given in percentages of removed cells.

### 2.4. Statistical Analysis

The Mann–Whitney U test was performed in SPSS software version 26 (IBM, Armonk, NY, USA) to confirm the statistical significance (*p* < 0.5) of differences among the crystal violet OD values of each experimental set of wells.

## 3. Results

### 3.1. Resistance to Disinfectants

In our study, minimal inhibitory concentrations (MICs) of three disinfectants were determined. While MIC values were 0.75% for Ecocid^®^ S and 9.0% for ethanol for all tested strains, MIC values for hydrogen peroxide were 0.015% (*n* = 12) and 0.029% (*n* = 3; strains 101/08, 134/19; 323/19), respectively.

### 3.2. Biofilm Formation

The in vitro biofilm-formation ability of strains was tested under various temperature conditions and incubation times (Figure 1). Regarding OD_595_ values, all tested strains formed biofilm on polystyrene at 20 °C and 28 °C after 72 h and 168 h incubation time, with a significant difference compared with negative controls. The results showed that the levels of biofilm were significantly higher (*p* < 0.001) at 20 °C and had a longer incubation time (168 h). Concerningly, the values of CV at 8 °C for some of the strains after 72 h (*n* = 6) and 168 h (*n* = 11) were significantly (*p* < 0.0.5) higher than those of negative control. Regarding the presence of pESI plasmid, the biofilm formation for strains carrying pESI plasmid was significantly higher at 20 °C (*p* < 0.001) and 28 °C (*p* = 0.003) after 168 h incubation time. However, there were no statistically significant differences in biofilm formation among the strains after 72 h, at any test temperature. The biofilm formation of 251/12 strain with the truncated pESI plasmid did not differ significantly from strains without the plasmid, and they are, therefore, grouped together. 

To determine the effect of temperature and incubation time, isolates were classified as strong, moderate, weak, and no biofilm formers (Table 2). After shorter incubation time (72 h), the majority of strains formed weak biofilm, regardless of temperature tested. On the contrary, the biofilm-forming ability after 168 h was classified as strong (*n* = 10 at 20 °C; *n* = 8 at 28 °C).

#### 3.2.1. Antiadhesion Effect of Disinfectants

Disinfectants in subinhibitory concentrations (1/8 MIC) were tested for their antiadhesion properties against *S.* Infantis 323/19 strain after 24 h, 72 h, and 168 h at 20 °C (Figure 2). The greatest effect was observed after 24 h incubation time when OD_595_ values after incubation with Ecocid^®^ S (*p* < 0.001) and ethanol (*p* = 0.029) were significantly lower than those of the positive control. Ecocid^®^ S was effective also after 72 h incubation time, but after 168 h, no effect was observed for any of the disinfectants tested. 

#### 3.2.2. Antibiofilm Effect of Disinfectants

Effect of disinfectants on the 72-h- and 168-h-old *S.* Infantis 323/19 biofilm formed at 20 °C on polystyrene was tested. The most effective were 1% Ecocid^®^ S and 70% ethanol, as there were no viable cells detected after the treatment of 72-h- or 168-h-old biofilm as early as after 15 min contact time. On the contrary, 1% hydrogen peroxide was effective on biofilm cells (no viable cells detected) only after a 30 min contact time. In the case of a 15 min contact time, the number of viable cells in the 72-h-old biofilm decreased by 50% and by only 18% after the treatment of the 168-h-old biofilm.

## 4. Discussion

Attachment and subsequent biofilm formation of food-borne pathogens have become one of the most important areas of food safety research [25]. In fact, biofilm formation is believed to enhance the capacity of pathogens, such as *Salmonella*, to survive under stressful environmental conditions and persist outside the host [26]. Moreover, biofilms formed in farm and food processing environments are a constant source of microbial contamination that may lead to infection or food spoilage. The biofilm-forming ability of *Salmonella* spp. on different abiotic surfaces, such as polystyrene, glass, plastic, cement, and rubber, has been confirmed [27]. The results of Moraes et al. (2018) indicated that most of *S. enterica* strains were characterized as moderate to strong biofilm producers. This phenotypic feature varied both among and within serovars and is observed mostly among strains isolated from food involved in outbreaks [28]. However, biofilm formation is affected by various environmental factors, e.g., temperature, contact time, surface material, presence of organic material, pH, water activity, etc. [28,29,30,31]. Among those, temperature is considered as one of the most important factors for bacterial growth and, thus, a critical control measure in the food-processing chain. In our study, the polystyrene microtiter plate method with CV staining was used for biofilm screening of *S.* Infantis isolates originating from poultry, food, and humans. The results showed that the biofilm formation was strain-specific, with an observed trend of increased biofilm formation at 20 °C and longer incubation time (168 h) (Figure 1; Table 2). Higher biofilm levels observed after longer incubation time (Figure 1) could be attributed to the downregulation of genes responsible for biofilm formation in early biofilm (72 h) and upregulated expression in the mature biofilm [32,33]. As shown before, isolate-to-isolate variability was observed, whereas *S.* Infantis produced strong biofilm on polystyrene at either 25 °C (48 h) or 15 °C (120 h) [34]. On the contrary, in the study of Schonewille et al. (2012), *S*. Infantis strains were poor biofilm producers (48 h) on polystyrene in general, independent of the incubation temperature (20 ± 1 or 37 ± 1 °C) [35]. Similarly, as shown by Pate et al. (2019), the *S.* Infantis persistence on broiler farms was related more to its widespread occurrence in the broiler production chain and ineffective disinfection protocols than to its ability to form biofilm [16]. However, the deviation from our results could be attributed to incubation time, with the results of both mentioned studies obtained after 48 h. Moreover, when evaluating the effect of incubation temperature on *S.* Infantis biofilm formation, a great inconstancy of results can be observed. For example, Borges et al. (2018) observed no significant differences in biofilm production of *S.* Infantis at four evaluated temperatures (37 °C, 28 °C, 12 °C, and 3 °C) [36], whereas Karaca et al. (2013) showed that 20 °C is the optimum temperature for *Salmonella* biofilm formation [37]. However, it is generally accepted that a lower temperature reduces or even stops the bacterial enzymatic activity. Consequently, metabolism, reproduction, and adhesion to surfaces could be minimized or prevented [30,38]. Further research has shown that lowering the environmental temperature at processing premises decreases but does not prevent biofilm formation. In addition, the initial interaction of cells with abiotic surfaces may be even better at low temperatures, increasing their adhesion and biofilm formation [26,39]. In our study, the OD_595_ values at 8 °C after both incubation times were significantly higher than those of the negative controls for the majority of tested isolates. This confirms that *S.* Infantis is able to produce biofilm even at relatively low temperatures [28,36] and possesses a great risk for cross-contamination during food processing. These results are relevant considering that chilling is required for all raw product unless they are moved directly from the slaughter line to heat processing or cooking. It is well established that a direct link exists among contamination in food-processing environments, bacterial biofilms, and contamination of the end food products [40,41]. Therefore, *S. enterica* strains with biofilm-forming ability are expected to pose a relatively high food safety risk because of biofilm-released cells that may result in recontamination of foods during processing. On the contrary, as predicted by MacKenzie et al. (2017), host adaptation relieves the selection pressure placed on *Salmonella* to persist in environments and correlates with loss of the biofilm phenotype [42]. The study results show that the experimental conditions used were less biofilm-favourable for isolates of human origin, as after 72 h incubation time, most strains produced no biofilm, regardless of temperature (Table 2). In addition, the human *S.* Infantis isolates tested were not closely related to the Slovenian broiler isolates, as noted in a previous study [3]. A different source of *S.* Infantis for human infections was suggested, and it seems that the ability to form a biofilm is not as crucial as it is in the farm environment. As also reported by Harrell et al. (2020), there is no evidence that would implicate the involvement of biofilms in persistent and/or recurrent non-typhoidal *Salmonella* human infections [43]. 

The presence of pESI plasmid contributes significantly to the antimicrobial resistance and the pathogenicity of its carrier. However, the pESI plasmid carries not only resistance genes but also several virulence factors, among which a superior biofilm formation was observed [4]. Recently, an *S.* Infantis strain-specific infectivity in broilers was also shown for the first time, and an increased potential of pESI-positive strains to colonize and infect birds was observed. As broilers infected with the pESI-positive strain were major shedders, a superior biofilm could also be attributed to environmental persistence of strains in broiler houses and consistent reinfection of birds [44]. The majority of strains (*n* = 11) in our study harboured pESI-like plasmid, which is also a predominant *S*. Infantis clone in Slovenian broilers and humans [3]. Along with strain-specific biofilm formation, a significantly higher biofilm formation after 168 h incubation time at 20 °C and 28 °C was shown for strains harbouring pESI-like plasmid, whereas there was no difference between the strains after shorter incubation time (72 h) (Figure 1). Because of the specific plasmid-borne genes which enhance colonization, virulence, and fitness behaviour of the strains, it was suggested that once the plasmid is acquired by *S.* Infantis, it would likely be spread [45]. Thus, proper intervention strategies are needed to prevent further dissemination of pESI-positive strains along the food chain. Interestingly, among human isolates, no/weak biofilm was observed for pESI-positive 61/21 strain, while 321/20 strain without p-ESI like plasmid was a strong biofilm former (Table 2). This may indicate the involvement of other uncharacterized genes in biofilm formation, which will be the focus of our future studies.

In practice, different cleaning methods and disinfectants are used to eliminate food-borne pathogens from poultry farm environments and food-processing chains. However, increased resistance of mature *Salmonella* biofilm to disinfectants and antibiotics has been reported [13,21]. In addition, a variable efficiency of commercial disinfectant products is affected by several factors, including temperature, contact time, and carrier material [46]. In our study, three disinfectants were tested for their antiadhesion and antibiofilm effect. Although 1/8 MIC values of disinfectants were used, there was a significant reduction of *S.* Infantis 323/19 biofilm after 24 h incubation with Ecocid^®^ S and ethanol, measured as OD_595_ of bounded CV (Figure 2). After 72 h incubation time, only one of the disinfectants (Ecocid^®^ S) was effective, whereas no effect was observed after 168 h. The efficacy and safety of Ecocid^®^ S under both laboratory and practical clinical conditions has been shown before [47,48]. This disinfectant, with a potassium peroxysulphate as an active substance, acts as a powerful oxidation agent. The highest antiadhesion and antibiofilm potential of Ecocid^®^ S in this study could be attributed to its multicomponent composition that allows for good contact with the cell surface. However, an antiadhesion effect of disinfectants in sub-inhibitory concentrations was observed only after a shorter incubation time, emphasizing the importance of subsequent biofilm formation despite the disinfectant residue. Nevertheless, it was even shown that ethanol in sub-inhibitory concentrations was able to stimulate biofilm formation of *S.* Enteritidis on polystyrene (25 °C for 5 h) [49]. Therefore, the pros and cons of disinfectants residues in farm and food environment should also be evaluated regarding the biofilm formation. Although the effect of only three disinfectants on the viability of biofilm cells was tested in this study, there was a difference in effectiveness shown as a percentage of viable cells after the treatment of the 72-h- and 168-h-old biofilm. Two of the disinfectants (Ecocid^®^ S and ethanol) reduced cell viability as early as after 15 min contact time, whereas hydrogen peroxide was ineffective on the 168-h-old biofilm. This is consistent with previous studies reporting a lower efficacy against mature biofilms. Although it is well known that organic material in poultry farms or slaughterhouses decreases the efficacy of disinfectants, further studies are needed to test the in vitro efficacy of disinfectants on *S.* Infantis strains. Moreover, as shown by Drauch et al. (2020), disinfectants vary in their efficacy on *S.* Infantis strains, while a difference in organic matter burden (low vs. high protein concentration) can also influence the level of effectiveness of disinfectants [14].

To sum up, our study results suggest that *S.* Infantis isolates differ in their attachment depending on the temperature and incubation time, which may also influence their persistence in the processing environment. Furthermore, the inconsistency of experimental conditions makes it difficult to compare the results of studies that evaluated the effects of incubation time, temperature, and disinfectants on biofilm formation. Based on our results, study designs should test prolonged incubation times (at least 168 h) to get representative results on *Salmonella* biofilm formation. In addition, biofilm formation should be tested under farm- or slaughterhouse-simulating conditions, using surfaces from practice and applying, for example, bovine serum albumin to mimic organic residues on surfaces. The imitation of conditions in practice is also one of the main limitations of the present study. Due to the presence of genes involved in resistance to quaternary ammonium compounds (*qacE*Δ*1*) in pESI-positive strains, further studies are planned to examine detailed resistance of biofilm to this class of disinfectants used in the food production chain.

## 5. Conclusions

*S.* Infantis isolates from poultry, food, and humans produce biofilms on polystyrene surfaces with a higher potential at 20 °C and prolonged incubation time. Importantly, a significantly higher biofilm has been shown for strains harbouring pESI plasmid. Our study and literature-based results suggest that *S.* Infantis attachment and biofilm production depend on the encountered conditions, which may influence their persistence in the environment. Despite stringent cleaning and disinfection measures, recurrent infections of poultry flocks and cross-contamination in meat-processing chains are often reported. Therefore, more effective precautionary measures should be implemented to ensure biofilm prevention and removal in order to control the *S.* Infantis occurrence. However, the specific biofilm formation ability and differences in antibiofilm properties of the tested disinfectants show the need for further investigations and use of standardized test protocols.

## Figures and Tables

**Figure 1 microorganisms-11-00301-f001:**
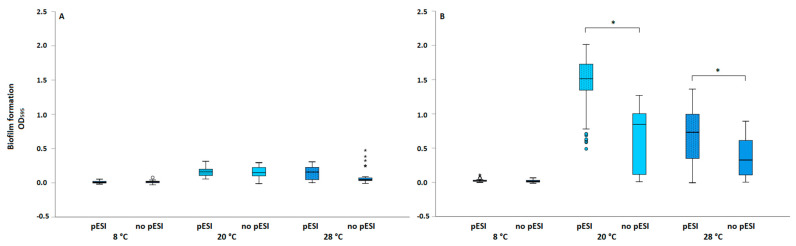
Biofilm formation of *S.* Infantis strains (*n* = 15) regarding presence of pESI plasmid, under various temperature conditions and incubation times: (**A**) 72 h and (**B**) 168 h; **⋆**, **○** outliers; * *p* < 0.05.

**Figure 2 microorganisms-11-00301-f002:**
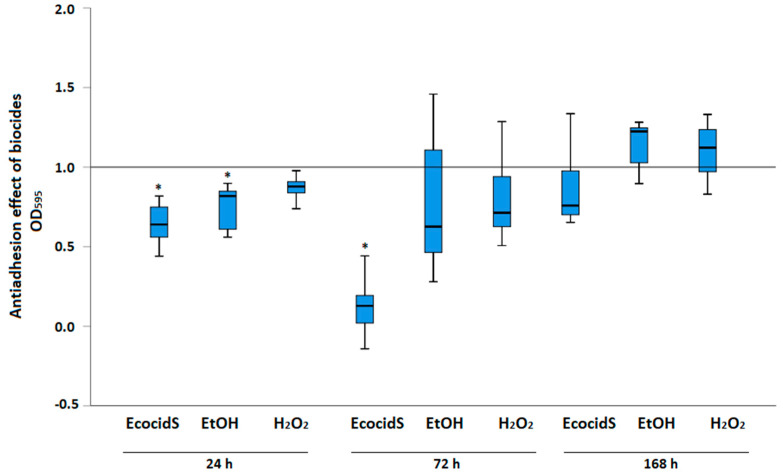
Antiadhesion effect of disinfectants on *S.* Infantis 323/19 strain after different incubation times at 20 °C. EtOH – ethanol; H_2_O_2_ – hydrogen peroxide; * *p* < 0.05.

**Table 1 microorganisms-11-00301-t001:** Metadata and characteristics of *Salmonella* Infantis strains used in the study (*n* = 15).

Strain Name	Isolation Year	Origin	pESI Presence ^1^	Resistotype (Coded) ^2^
101/08	2008	Broiler faeces	−	1
160/12	2012	Broiler faeces	+	2
251/12	2012	Broiler faeces	+ (truncated)	3
42/14	2014	Breeding flock	−	1
290/18	2018	Broiler faeces	+	2
42/19	2019	Food	+	2
134/19	2019	Food	+	2
219/19	2019	Broiler faeces	+	4
236/19	2019	Food	+	5
323/19	2019	Broiler faeces	+	2
321/20	2020	Human urine	−	1
58/21	2010	Human faeces	−	6
60/21	2013	Human faeces	+	2
61/21	2015	Human faeces	+	7
62/21	2019	Human faeces	+	8

^1^ pESI presence is indicated with a plus sign, absence with a minus sign. ^2^ Resistotypes are numbered as follows—1: *aac(6’)-Iaa*, *parC* (T57S); 2: *tet*(A), *aac(6’)-Iaa*, *aadA1*, *sul1*, *parC* (T57S), *gyrA* (S83Y), *nfsA* (NS159); 3: *aac(6’)-Iaa*, *parC* (T57S), *gyrA* (S83Y), *nfsA* (NS159); 4: *tet*(A), *aac(6’)-Iaa*, *aadA1*, *aadA2*, *sul1*, *sul3*, *dfrA8*, *cmlA1*, *parC* (T57S), *gyrA* (S83Y), *bla*_TEM-1B_, *nfsA* (NS159); 5: *tet*(A), *aac(6’)-Iaa*, *aadA1*, *sul1*, *parC* (T57S), *gyrA* (S83Y), *qnrS1*, *bla*_TEM-32_, *nfsA* (NS159); 6: *tet*(A), *aac(6’)-Iaa*, *ant(2’’)-Ia*, *sul1*, *dfrA1*, *cmlA1*, *parC* (T57S), *gyrA* (D87Y), *bla*_TEM-1B_; 7: *tet*(A), *aac(6’)-Iaa*, *aadA1*, *sul1*, *dfrA14*, *parC* (T57S), *gyrA* (S83Y), *nfsA* (NS159); and 8: *tet*(A), *aac(6’)-Iaa*, *aadA1*, *aph(3’)-Ia*, *aph(3’’)-Ib*, *aph(6)-Id*, *sul1*, *sul2*, *dfrA14*, *floR*, *parC* (T57S), *gyrA* (S83Y), *bla*_TEM-1A_, *nfsA* (NS159).

**Table 2 microorganisms-11-00301-t002:** Biofilm-forming ability of *S.* Infantis isolates on polystyrene at two temperatures and incubation times.

Temperature	20 °C		28 °C	
Strain Name	72 h	168 h	72 h	168 h
101/08	weak	weak	moderate	weak
160/12	weak	moderate	/	/
251/12	weak	weak	weak	weak
42/14	/	/	/	/
290/18	weak	strong	weak	strong
42/19	weak	strong	weak	strong
134/19	weak	strong	weak	strong
219/19	weak	strong	weak	strong
236/19	moderate	strong	weak	strong
323/19	weak	strong	moderate	strong
321/20	/	strong	weak	strong
58/21	weak	strong	weak	moderate
60/21	/	strong	/	moderate
61/21	/	weak	/	weak
62/21	moderate	strong	weak	strong

Biofilm formation was classified using the cut-off value (ODc) as strong biofilm: 4× ODc < OD, moderate biofilm: 2× ODc < OD ≤ 4× ODc, weak biofilm: ODc < OD ≤ 2× ODc, and no biofilm (/): OD ≤ ODc.

## Data Availability

The data presented in this study are available on request from the corresponding author.

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
