# Peer review of "Antimicrobial and Antibiofilm Effect of Commonly Used Disinfectants on *Salmonella* Infantis Isolates"

_microorganisms, 2023, doi:10.3390/microorganisms11020301_

Round 1
Reviewer 1 Report
The presented paper is written excellently, all the methods and results are presented in such manner that they allow complete reproduction and understanding of the topic. I especially salute the fact that the authors pointed out the local (Slovenia) significance of Salmonella strains in the light of given topic.
My only suggestion for changes- please add the composition of Ecocid, as some reader may ask what is the active principle in this disinfectant. Potentially, make additional comment on its mechanism of action and/or previous experiences with this disinfectant.
Author Response
REVIEWER 1
Point 1: The presented paper is written excellently, all the methods and results are presented in such manner that they allow complete reproduction and understanding of the topic. I especially salute the fact that the authors pointed out the local (Slovenia) significance of Salmonella strains in the light of given topic.
Response 1: We appreciate very much for review of our paper.
Point 2: My only suggestion for changes- please add the composition of Ecocid, as some reader may ask what is the active principle in this disinfectant. Potentially, make additional comment on its mechanism of action and/or previous experiences with this disinfectant.
Response 2 : The composition of Ecocid was added in the Materials and Methods paragraph as follows (Line 112): “To determine the in vitro antimicrobial effect of Ecocid® S (a mixture of peroxy compound, surfactant, organic acids, and inorganic buffer; Krka, d.d., Novo mesto, Slovenia), ethanol (Sigma-Aldrich, Darmstadt, Germany) and hydrogen peroxide (Sigma-Aldrich, Darmstadt, Germany) the broth microdilution method was used.”
Furthermore, a comment on its possible mode of action and previous research was added in the Discussion paragraph as follows (Line 331): “The efficacy and safety of Ecocid® S under both, laboratory and practical clinical conditions, has been shown before [46,47]. This disinfectant with a potassium peroxysulphate as an active substance acts as a powerful oxidation agent. The highest antiadhesion and antibiofilm potential of Ecocid® S in this study could be attributed to its multi-component composition that allows good contact with the cell surface.”

Reviewer 2 Report
This a well-written article.
Line 21- The authors have mentioned “significantly higher”, please add your p-value within parentheses.
Line 116- Why didn’t you use Mueller Hinton broth instead? Did you follow any standards for antimicrobial assay? What are the ranges of concentrations of the disinfectants used for broth dilution?
Line 155 – were prepared
Line 178 – Do statistical tests represent just the biofilm-forming ability? What about the other assays?
Line 246 – Use Oxford comma in the list, before “and” and what is the meaning of …?
Author Response
Point 1: Line 21- The authors have mentioned “significantly higher”, please add your p-value within parentheses.
Response 1: Regarding the comparability of the interpretation of the results we decided to let out the “significantly”. However, the level of significance was added in the Results section (Line 194). The latter is applied to the difference in OD595 values cumulatively for all strains that were significantly higher at 20 °C (both incubation times 72 and 168 h) and longer incubation time (168 h) at all tested temperatures.
Point 2: Line 116- Why didn’t you use Mueller Hinton broth instead? Did you follow any standards for antimicrobial assay? What are the ranges of concentrations of the disinfectants used for broth dilution?
Response 2: The aim of testing the microbial resistance to disinfectants was to set the MIC (minimal inhibitory concentration) values in order to determine the antiadhesion and antibiofilm effect of disinfectants. In preliminary research, the highest biofilm potential was shown in TSB medium. For this purpose, Tryptone Soya Broth (TSA) was used for all of the experiments.
For better transparency, a reference was added in the Materials and methods section (Line 116). Although the most common method of MIC determination is based on optical density measurements, we used PrestoBlue™ Cell Viability Reagent in order to avoid false positive results.
Point 3: Line 155 – were prepared
Response 3: Was corrected.
Point 4: Line 178 – Do statistical tests represent just the biofilm-forming ability? What about the other assays?
Response 4: The statistical analysis was performed only for the determination of differences in OD595 values for the results of Biofilm formation (Paragraph 3.2) and the Antiadhesion effect of disinfectants (Paragraph 3.2.1).
Point 5: Line 246 – Use Oxford comma in the list, before “and” and what is the meaning of …?
Response 5: An Oxford comma was added.
The meaning of three dots (…) is referring to other factors that may influence the biofilm formation. It was replaced by “etc.”
Additionally, the spell check of the article was re-done and we believe there is an improvement.
